# Micronutrients at Supplemental Levels, Tight Junctions and Epithelial Barrier Function: A Narrative Review

**DOI:** 10.3390/ijms25063452

**Published:** 2024-03-19

**Authors:** Katherine M. DiGuilio, Elizabeth A. Del Rio, Ronald N. Harty, James M. Mullin

**Affiliations:** 1Lankenau Institute for Medical Research, 100 Lancaster Avenue, Wynnewood, PA 19096, USA; diguiliok@mlhs.org (K.M.D.); delrioe@mlhs.org (E.A.D.R.); 2Department of Pathobiology, School of Veterinary Medicine, University of Pennsylvania, 3800 Spruce Street, Philadelphia, PA 19104, USA; rharty@vet.upenn.edu; 3Division of Gastroenterology, Lankenau Medical Center, 100 Lancaster Avenue, Wynnewood, PA 19096, USA

**Keywords:** tight junction, micronutrient, zinc, vitamin D, claudin, signal transduction, diet, evolution, inflammation, Tumor Necrosis Factor

## Abstract

Disease modifiers, whether from cancer, sepsis, systemic inflammation, or microbial pathogens, all appear to induce epithelial barrier leak, with induced changes of the Tight Junctional (TJ) complex being pivotal to the process. This leak—and the ensuant breakdown of compartmentation—plays a central role in disease morbidity on many levels. Accumulation of lung water in the luminal compartment of airways was a major driver of morbidity and mortality in COVID-19 and is an excellent example of the phenomenon. Increasing awareness of the ability of micronutrients to improve basal barrier function and reduce barrier compromise in pathophysiology may prove to be a low-cost, safe, and easily administered prophylactic and/or therapeutic option amenable to large populations. The growing appreciation of the clinical utility of supplemental doses of Vitamin D in COVID-19 is but one example. This narrative review is intended to propose a general theory on how and why micronutrients—at levels above normal dietary intake—successfully remodel TJs and improve barrier function. It discusses the key difference between dietary/Recommended Daily Allowance (RDA) levels of micronutrients versus supplemental levels, and why the latter are needed in disease situations. It advances a hypothesis for why signal transduction regulation of barrier function may require these higher supplemental doses to achieve the TJ remodeling and other barrier element changes that are clinically beneficial.

## 1. The Singular Importance of Fluid Compartmentation and Epithelial Barriers

In any higher organism as complex or more complex than coelenterates, the critical importance of barrier function and compartmentation for homeostasis is an obvious reality. Fundamentally, humans are parallel arrays of fluid compartments delimited by typically single-cell layers of epithelia. This organization holds for nearly every one of our organs. In nearly every instance, a tissue-specific luminal compartment is separated from a common anti-luminal compartment (the blood stream) by the tissue-specific epithelium.

That ramified architectural reality is true for the lung, liver, nephron, choroid plexus, testis, uterus, the entire gastrointestinal (GI) tract, and any gland in the body. Below, we will briefly describe how nearly every disease pathology seeks to subvert that barrier function, thereby compromising fluid compartmentation. Notably, this assault on barrier function has two very general outcomes for disease etiology and morbidity. The first is that microbial pathogens (which almost universally seek systemic access to us by first encountering the apical surface of an epithelial cell layer) will either actively compromise that epithelial barrier or take advantage of an already compromised barrier to systemically infect an organism [1,2,3]. This is as true for *Clostridium perfringens* (where a bacterial pathogen infects and transits an initially healthy barrier) [4] as it is for celiac disease (where the protein, gluten, and the peptide, gliadin, transit the barrier) [5,6]. Second, aside from pathogens and pathogenic substances crossing a barrier, there is another consideration which figures enormously in disease morbidity. Epithelial barriers are interfaces for thermodynamic work—the most essential aspect of our systemic physiology. Through the process of energy-dependent active transport, polar epithelial cells translocate a wide array of solutes against their chemical or electrochemical gradients at the expense of ATP. If there is a (paracellular) compromise of the barrier, it will short-circuit that work, resulting in both a fluid dynamic as well as a bioenergetic downside. For example, in infectious diseases, compromise of the GI barrier, specifically, not only facilitates microbe and immunogen access to the interstitium (septic infection/inflammation), but it also creates a fundamental condition for solute and fluid leakage (diarrhea and its physiological downside outcomes such as systemic potassium loss). There is also an often-overlooked potential bioenergetic cost in the futile cycle of ATP consumption from transcellular transport, paracellular back leak, and repeated cycles of the same. One does not need to look any further than the generation and medical significance of lung water in COVID-19 victims as recent, dramatic proof of the clinical importance of barriers and the subversion of these highly fundamental transepithelial transport processes. This “pump/leak” phenomenon for epithelia has long been recognized, along with the role of ATP, whether the “leak” component has been transcellular or paracellular [7,8].

The lynchpin in this entire scenario is the TJ complex. As described in several review articles [9,10,11], the TJ complex is just that—complex. Approximately 30 integral membrane proteins can form the barrier itself, most notable being the 27-member (23-gene) claudin family. But, in addition, there are at least 10 TJ-associated, intracellular proteins—both structural and signaling proteins—that regulate the complex and interface it with the cytoskeleton. Moreover, there is a growing literature that suggests that the core barrier proteins can have a signal transduction role of their own [12,13]. Due to this complexity, one realizes that producing a de novo drug (small molecule or protein) designed to alter and *improve* barrier function is an exceedingly difficult endeavor. Indeed, every disease process and pathogen seem adept at TJ remodeling that *compromises* barrier function. For this reason, it is, at first glance, highly remarkable that naturally occurring substances in the human diet—micronutrients—can *improve* barrier function and/or reduce its compromise in various disease situations (Figure 1).

While it is, on the one hand, fully remarkable that naturally occurring substances exist that remodel TJs in such a way that they are less—not more—leaky, there is another consideration that makes the phenomenon even more remarkable. The TJ is not the only element of importance in barrier function. One can create leak in a barrier simply by “shooting holes” in it by killing cells [14]. Necrosis, apoptosis, autophagy are ever present phenomena in all epithelial barriers, and are increasingly recognized as playing a role in barrier function, sometimes even (counterintuitively) a positive one [15]. It is *how* these processes occur within a barrier that determines whether they are deleterious or not [16]. This makes the barrier-supportive action of micronutrients even more impressive because while they beneficially remodel TJ complexes, these micronutrients must not negatively interfere with apoptosis, autophagy (and its attendant cellular remodeling) or even mucus secretion, which also play key roles in any barrier.

## 2. The near Universal Occurrence of Barrier Leak in Disease

While a colon perforation during colonoscopy is an undeniably adept means of creating gastrointestinal barrier leak, nature—specifically, pathophysiological nature—has evolved much more subtle and finessed approaches at creating leak. In active inflammatory bowel disease (IBD), actual lesioned erosions of the GI mucosa certainly result in leak. However, well before these gross manifestations of the disease, molecular alterations occur that increase leak, and these are focused on the TJ complex. The evidence for IBD-related TJ alterations is now myriad [17,18,19]. Related to, and partially underpinning, these IBD-induced TJ changes is a witch’s brew of proinflammatory cytokines that frequently form the disease vanguard of leak creation. Proinflammatory cytokines such as Tumor Necrosis Factor-alpha (TNF-α) have been extremely well-identified as TJ leak-inducing agents in all investigational models available: cell culture, animal, and human, and they do so in a wide array of epithelial tissues [20,21,22,23]. And since cytokines seem to hunt in “packs”, it is worth noting that combinations of proinflammatory cytokines can not only exacerbate TJ leak, but also induce leak arising out of (unregulated) cell death in the barrier [14].

Microbial pathogens—both viruses and bacteria—are well-evolved to induce leak as part of their etiology [24]. Examples of viruses would include rotaviruses, flaviviruses, influenza viruses and coronaviruses, to name just a few. On the bacterial side, examples include *Clostridium perfringens*, *Neisseria meningitidis* and *Campylobacter jejuni*, again to name a few. Among both viral and bacterial pathogens, there exists a subcategory worth mentioning. Some pathogens have evolved the ability not only to induce leak as part of their etiology, but also to target the TJ complex specifically. For example, the McClane group has produced a large body of work showing that *Clostridium perfringens* releases an enterotoxin that binds to claudins as a means of cellular invasion and paracellular barrier disruption [4,25]. Hepatitis C virus utilizes claudin-1 in its entry into cells and in its induced changes in the TJ complex [26]. Influenza viruses have shown affinity for the PDZ domains of TJ-associated proteins [27]. The SARS-CoV-2 coronavirus was shown recently to have affinity for the PDZ domain of the TJ associated protein, ZO-1 [28]. When one considers that across the phylogenic spectrum, from viruses to bacteria to even dust mites (whose feces contain a protease with remarkable affinity for TJ barrier proteins [29]), the frankly amazing focus by pathogens on our epithelial barriers in general, and on our TJ complexes specifically, is perhaps the best existing testament for the singular importance of barrier function in health and disease.

However, it is not simply infectious diseases that target the TJ complex. TJ integrity has been a casualty in various cancers. For example, TJ leak has been demonstrated in early stage colon cancer [30] and in mammary and liver cancer, as well [31]. More fundamentally, it is worth noting that the tumor promoter class of carcinogens is very adept at inducing TJ leak through their activation of Protein Kinase C isoforms [32,33]. The luminal sequestration of specific growth factors and their leak into the interstitium in vivo may result in the barrier compromise being of great significance in cancer specifically [34]. As well-known examples of autoimmune diseases that achieve morbidity in part through induced TJ leak, celiac disease [5], Crohn’s disease and Ulcerative Colitis [35] figure prominently. TJ leakiness has additionally been associated with rheumatoid arthritis [36]. Less well-known is the occurrence of TJ leak in diabetes where leak in the blood–brain barrier (BBB), blood retinal barrier and GI tract have been observed [37,38,39].

To summarize, TJ leak is seen in many widely different diseases in a wide range of epithelial tissues, only a few of which are mentioned here. Given the unique TJ disruptions in those various diseases, as well as the uniqueness of TJ complexes in different epithelial tissues, clinically useful agents countering (or preventing) disease-induced leak would likely need to achieve that action in such a way as to at least be neutral concerning ALL aspects of barrier function (in addition to, but not exclusive to, TJ permeability). Likewise, such interventions would also need to be at least harmless for all unaffected epithelial tissues in the organism.

## 3. Micronutrient Reduction of Barrier Leak

A literature search using the PubMed search engine for the terms micronutrient/vitamin/flavonoid/zinc AND tight junction (the basis for this review and our previous, more exhaustive review [24]) now pulls in over 1000 peer-reviewed publications. Many excellent reviews have been written on the subject and it currently seems a well-established fact that micronutrients can improve basal epithelial barrier function [24,40,41,42]. In addition, micronutrients have demonstrated efficacy in reducing the barrier compromise caused by disease agents and modifiers [43,44,45]. In doing this, they have shown the promise of clinical value as both preventatives and therapeutics in various disease scenarios. Using COVID-19 as an example, it is instructive to consider the now-demonstrated utility of the micronutrient vitamin D3 in this disease’s severity [46,47,48]. The clinical picture here is well-supported by vitamin D’s reported ability to improve airway barrier function, as well as reduce the airway barrier compromise caused by proinflammatory TNF-α in in vitro studies [29,45]. It is interesting to consider whether a widespread use of a Vitamin D supplement at the height of the COVID-19 pandemic would have supported SARS-CoV-2-compromised airway barrier function and measurably reduced the disease’s toll on health care systems worldwide.

A common thread that runs through many of these published reports is that the micronutrients are showing effectiveness—in clinical situations as well as in cell culture models—at concentrations above what would be consumed through normal dietary intake (as well as official RDAs) (discussed in detail in DiGuilio et al., 2022) (Figure 2A). In the case of zinc, for example, normal dietary intake is in the range of 5–10 mg/adult/day. The toxic limit for daily zinc intake is approximately 150 mg/adult/day [49]. However, we would maintain that a “therapeutic range” exists from approximately 50–100 mg/day, wherein clinical effectiveness seems to exist regarding barrier function (Figure 2B).

This therapeutic intake will give a transient spike of blood zinc concentration of approximately 50–100 µM, which curiously mirrors the 50–100 µM concentration range shown to be effective in many in vitro studies with human epithelial cell culture models [50,51]. A similar situation seems to exist regarding micronutrients in general. The actual effective concentrations are unique to each micronutrient but the phenomenon of a “therapeutic range” above the RDA level seems valid. This is clinically significant because it means that micronutrient intake achieved through diet or through a general “multi” vitamin/mineral regimen, will be insufficient to induce the epithelial barrier effects discussed here.

Another important point is that there is pronounced tissue specificity in the micronutrient effects on barrier function. For example, zinc has been extensively demonstrated to be effective at improving GI barrier function in cell culture, animal, and human studies [50,52,53], but is rather ineffective in airway epithelial models [54]. Such tissue specificity would seem to translate to a need to consider the tissue/barrier most compromised in a specific disease (e.g., airways in COVID-19), and which micronutrient supplementation would be most effective. Again, one needs the micronutrient to be effective in, e.g., the airways, but not compromise barrier function in, e.g., the nephron, retina, or small bowel.

A final consideration concerns translational speed. In our own research group, we initiate micronutrient/barrier function studies on well-differentiated human epithelial cell culture models and then—if successful there—pivot to patient-based (human clinical) studies. We can do this in part because of the well-described safety profile for these dietary substances. In fact, most are accorded GRAS (Generally Recognized as Safe) status by the US Food and Drug Administration. However, what is a welcome property for clinical research studies becomes something very different and highly important on a mass population level in critical medical situations such as pandemics. These micronutrient substances can be rushed into clinical use faster than any drug, even repurposed drugs. During COVID-19, for example, there was no need for safety studies to utilize supplement levels of vitamin D in large populations. That fact, plus the relatively hearty stability of many micronutrients (zinc being the best example) and their amenability to oral administration, means relatively uncomplicated storage and field distribution—huge factors in rapid and widespread application.

## 4. The Case for Adjuvant Therapy

Too often in medical research, we exclusively seek to know how a disease can be cured or thoroughly prevented, as, e.g., Salk, Sabin, and polio. Indeed, we witnessed this in the very strong focus on the mRNA vaccines during the COVID-19 pandemic. But there is a very strong case to made for the value of supportive therapy, enabling the patient’s own immune system to cure the disease by simply maintaining the host’s physiology. We would advance that micronutrient effects on barrier function can be just that.

At the height of COVID-19, one’s ticket into an intensive care unit (ICU) (or a ventilator) was the development of lung water, a very good example of the barrier leak being discussed here. Numerous in vitro studies have shown vitamin D’s ability to improve barrier function and to temper the barrier leak that ensues with a “cytokine storm”, such as the one that played such a pivotal role in COVID-19 mortality [55,56,57]. In hindsight, we are now realizing the high value that vitamin D therapy could have played in COVID-19 [46].

Would this micronutrient have prevented the disease? No. Cured the disease? Again, no. But through its supportive action on airway barrier function, it very likely would have mitigated the course of the disease, reducing ICU and ventilator census at a highly critical time. Furthermore, in the case of COVID-19, we do not even know if vitamin D would have been the *best* choice. For example, the flavonoid micronutrient, quercetin, also shows remarkable ability to support airway barrier function, as does vitamin A [58,59]. These considerations formed the basis of a very recent review [24]. Clearly, much more research in this area is needed.

No one would likely only recommend a micronutrient therapy for a diseased patient and then cavalierly wish them luck. Clearly, other pharmacologic and/or immunological approaches are merited in any multifaceted defense against a disease. But to eschew the micronutrient approach because it only partially succeeds would conversely seem equally reckless. The question may devolve to, what is a 15% improvement in morbidity worth? And the answer to that question needs discussion on both the level of an individual patient as well as an at-risk population.

## 5. A Hypothesis for How Micronutrients at Elevated Levels Are Effective in Supporting Barrier Function

In considering the question of the efficacy of specifically *elevated* levels of micronutrients in supporting barrier function, it is instructive to consider the role of zinc in Protein Kinase C (PKC) activity. It has been known for quite some time that zinc can bind to PKC, with X-ray fluorescence indicating four zinc ions per molecule [60]. Zinc was observed to interact with PKC in a Ca^2+^-dependent manner that can be stimulatory or inhibitory, depending upon the level of calcium [61]. This action of zinc extends not simply to PKC *activity*, but also affects its translocation, including translocation to the cytoskeleton [62,63]. Of note for the issue of zinc as a micronutrient is that zinc is achieving this PKC regulation at concentrations centering around 50 µM [61,62,63,64]. This concentration happens to coincide with the levels at which zinc is reported to show efficacy in epithelial barrier function in cell culture models [50,51]. This, in turn, correlates with the supplemental levels of zinc showing efficacy in clinical studies regarding barrier function [52], with dosages of zinc in the range of 100 mg, and distinctly above the dietary/RDA level. We mention the example of zinc and PKC not because PKC is a known regulator of TJ permeability [65,66] or that PKC is an intracellular “TJ associated protein” believed to be physically associated with the PAR3 and PAR6 proteins [9], but simply as an example of a micronutrient regulating a signaling protein and doing so at concentrations that would not be achievable with normal dietary intake (but fully achievable with supplement dosages). The overall situation is not isolated simply to zinc and PKC. Supplemental levels of Vitamin D3 can activate Protein Kinase A in direct, non-genomic action [67].

In considering the issue of dosage, it is useful to invoke some very basic enzyme kinetics. Figure 3 illustrates the standard “Michaelis Menten” relationship of substrate concentration and enzyme activity. Even though we are discussing a cofactor (zinc) and a signaling protein (PKC), the same relationship holds for zinc concentration and PKC activity (or level of inhibition). The v vs. [S] plot essentially defines the relationship. We theorize that a hypothetical Protein Kinase X with a high affinity for zinc has half-maximal saturation/activity at the normal dietary blood level at 5 µM of zinc. Protein Kinase Y, on the other hand, has much lower affinity for zinc and does not achieve half-maximal activity until 50 µM zinc (Figure 3). In this scenario Protein Kinase Y would only exhibit significant activity with the *supplemental* level of zinc that results in a 50 µM or greater blood concentration.

Figure 4 and Figure 5 attempt to place these considerations of micronutrient concentration, signaling enzyme kinetics and TJ regulation, in the overall (highly convoluted) context of intracellular signal transduction with its myriad interconnecting pathways. Whereas a 5 µM zinc concentration achieves only partial activation of signaling pathways regulating a TJ complex (Figure 4A red pathway [Protein Kinase X]), a 50–100 µM concentration achieves activation of both Protein Kinase X *and* Protein Kinase Y (Figure 4B, red and orange pathways), which results in additional signaling activation that then remodels the TJ complex to become less leaky.

A “real life” example of this phenomenon concerns the ERK signaling pathway and is somewhat more complicated. The TJ remodeling and improved barrier function reported to occur with elevated levels of micronutrients is often associated with *reduced* ERK activity and lower levels of pERK. This was reported for the micronutrient Quercetin in airway epithelia [59]. It has also been reported for zinc in intestinal epithelia [68]. So, in this instance, one has an elevated level of a micronutrient such as zinc turning *down*/*off* a signaling pathway, which then leads to TJ remodeling with resultant reduced TJ leak. This is shown in Figure 5.

Yet, the question remains: why do our signaling pathways (and TJ complexes) respond favorably to higher than dietary levels of a micronutrient? Again, using zinc and PKC as examples, what if, over the millennia, a particular PKC isoform that began its existence as a signaling protein with high affinity (5 µM Km) for zinc (Protein Kinase X) evolved through accumulated polymorphisms on at least one allele to become Protein Kinase Y with reduced affinity for zinc? In this case, only (heterozygous) individuals who consumed a high dietary intake of, in this case, zinc would have an active Protein Kinase X and Protein Kinase Y, and their resulting downstream functional results would lead to fully favorable TJ remodeling in disease states. If the dietary intake was high, one would still be benefitting from an active Protein Kinase Y, even though one had accumulated, over time, a series of Protein Kinase Y mutations (polymorphisms) that reduced its affinity for zinc.

The human race was not nearly as mobile 100,000 years ago (even 100 years ago) as it is today. And this had profound implications for diet and nutritional intake. If one happened to live in a certain valley where life was reasonably good and food was plentiful, one tended to stay put. That also meant that one ate what was nearby, and, thus, one’s diet could be somewhat monotonous. Going back to prehistory, it has been noted that those human settlements situated along marine/aquatic food sources tended to be more geographically sedentary, implying that they ate what was nearby, and that the lower socioeconomic classes of those societies (where most of the gene pool would reside due to the greater population size) consumed the least diverse, most monotonous diet [69,70]. That monotony might have ensured high levels of specific micronutrients deriving from that diet. If one lived near a salmon-filled lake or stream, one ate a great deal of salmon. If that valley also happened to contain plentiful supplies of mushrooms, one would consume a lot of those as well. A diet high in certain mushrooms and salmon would make for a diet very high in the micronutrients zinc and vitamin D. And, thus, when one encountered a disease situation calling for enhanced barrier function (deriving from high intake of these micronutrients) one’s epithelial cell layers would be ready. *However*, remove one from that high and chronic intake of specific micronutrients, and Protein Kinase Y’s high Km shortfalls would become very apparent. Therefore, the need for micronutrient supplementation in specific disease situations would be born.

## 6. The Economic Realities

Given their natural availability, it is difficult to secure a patent on zinc (or vitamin D, etc.). One can generate a “Use Patent” for a particular application or utility of a micronutrient, but these tend to be more problematic than the more defensible, compositional patents such as those for a novel drug formulation. In the context of why micronutrient research may not be the darling of research institutions worldwide, the issue is even more compelling in the sphere of the pharmaceutical/biotech industry. Again, how much can one charge for zinc? A little creativity here might solve that issue, but it is an uphill struggle in terms of bureaucracy. For example, the fish oil (omega-3 fatty acid) supplement industry has been long saddled with the issue of purity, specifically purity from contaminating toxic levels of mercury. So, similarly for zinc, how much would the public pay for a source of supplemental zinc that was truly free of other divalent heavy metals such as cadmium (especially given the need for supplements to sometimes be taken on a daily, long-term basis). So, a profit element is not fully alien to the micronutrient “industry”. Returning to the example of COVID-19, where might one invest one’s Research and Development outlay? In a micronutrient for which one could only charge dimes, or an mRNA vaccine or antibody therapy for which one could charge thousands of dollars for individual dosing?

COVID-19 will not be our last pandemic, and one must ask, how many hyper-expensive therapies can the world afford? Moreover, will those therapies be available to mass populations, given their cost? If future pandemics involve pathogens worse than coronaviruses, one might need to make serious use of low cost, low tech, rapidly, and easily administered therapeutics, and micronutrients are thankfully at the world’s disposal.

## 7. Conclusions

Micronutrients will never be dramatic game changers in therapeutic medicine. They will not fully cure and they will not fully prevent. Still, they do address an extremely fundamental component of disease—morbidity arising from the leak that is induced across epithelial tissue layers, a phenomenon that encompasses virtually all types of pathobiology. In addition, although they may not cure, they can significantly reduce disease morbidity at a very fundamental plane, thereby potentially alleviating pressure on health services in public health crises. Moreover, they are relatively safe, able to be administered orally, often chemically stable, and available at very low cost compared to de novo small-molecule drugs or biologics, an important consideration in national public health budgets. The co-evolution of humans with dietary micronutrients has aided their medical functionality. Their utility in disease situations at levels above typical dietary intakes may follow in part from this co-evolution and may give rise to the reassessment of the RDA.

## Figures and Tables

**Figure 1 ijms-25-03452-f001:**
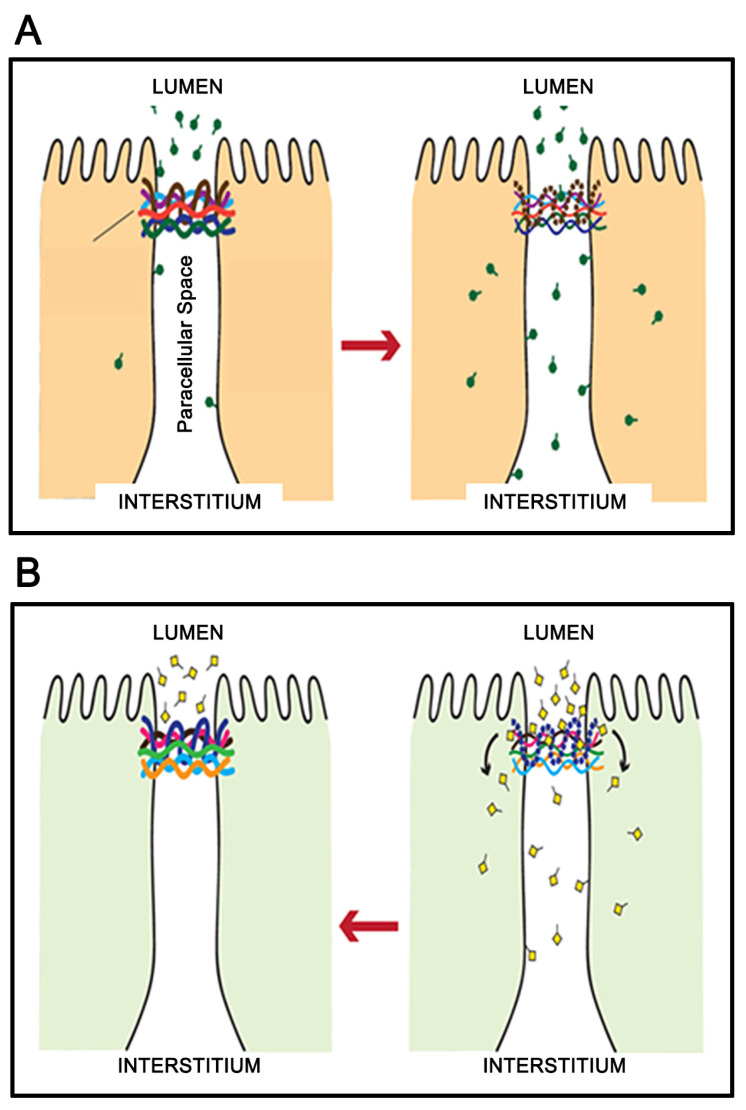
Disease Modifier-Induced Barrier Leak vs. Micronutrient-Induced Barrier Tightening. A very diverse array of diseases induces tight junctional (TJ) remodeling, resulting in increased TJ permeability (leak) (**A**). Such leak can, in certain cases, allow for the penetration of pathogens, but also short circuits thermodynamic work performed by the polar epithelial cells. Select micronutrients can, however, induce a different, beneficial TJ remodeling that results in less leak basally and/or decreased ability of the disease modifiers to induce a leak (**B**).

**Figure 2 ijms-25-03452-f002:**
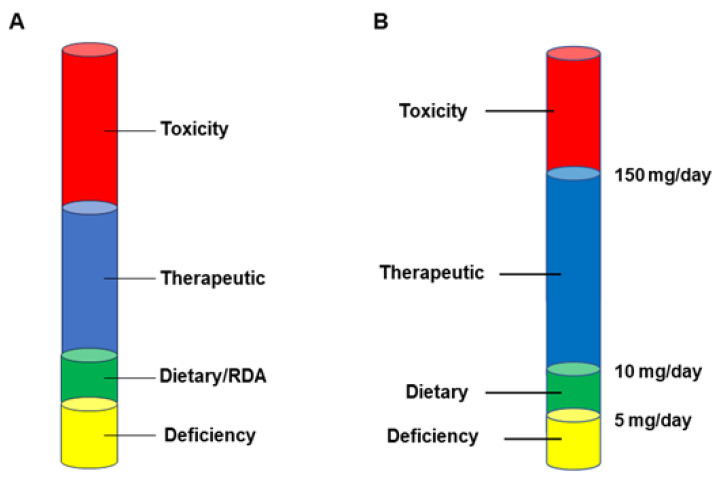
The Importance of Dosage in Micronutrient Action. For every micronutrient, there is a wide concentration range regarding its functionality concerning physiology and disease. Defining the range is the amount of the micronutrient taken in from a normal healthy diet, what could be termed the dietary or Recommended Daily Allowance (RDA) quantity. Below this range, one would be considered deficient for that micronutrient. Above that range, there will be an amount where toxic effects of that micronutrient will manifest. But between that toxicity threshold and the amount taken in from (normal) dietary sources, there will be a “therapeutic range” where beneficial effects can occur that are not manifested from normal dietary intake, and yet be well below toxic levels (**A**). Zinc represents possibly the most thoroughly researched micronutrient regarding barrier function, and its toxicity limit of approximately 150 mg/adult/day has been long known. In addition, normal dietary intake (non-vegan diet) is well-known to contribute 5–10 mg of zinc per day. An intake below 5 mg is typically considered to be a deficient diet. In a range around 50–100 mg/adult/day, therapeutic effects of zinc in various disease scenarios have been reported (**B**).

**Figure 3 ijms-25-03452-f003:**
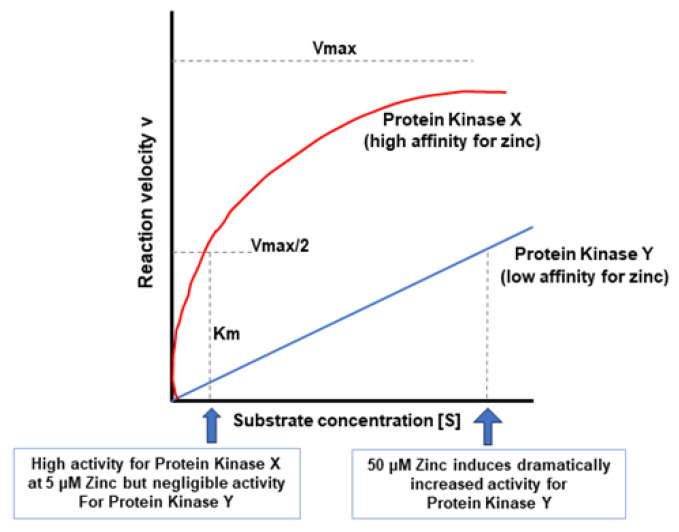
Basic Michaelis Menten Kinetics Considerations for Zinc and Signaling Proteins. The kinetic relationship between substrate concentration and reaction velocity that defines all enzymes also holds true for signaling proteins. The Km value particularly (the substrate [or cofactor] concentration that results in a half-maximal reaction velocity), is a defining characteristic for that protein. The higher the Km value, the lower the affinity of the protein for the substrate or cofactor. A signaling protein kinase (Y) with a 50 µM Km for zinc will not have significant activity at the 5 µM zinc (blood) concentration that accrues from normal zinc dietary uptake. However, a high affinity protein kinase (X) will have significant activity. A 5 µM zinc concentration will produce downstream effects of Protein Kinase X but not of Protein Kinase Y. A 50 µM zinc concentration would be needed to bring both signaling proteins into play.

**Figure 4 ijms-25-03452-f004:**
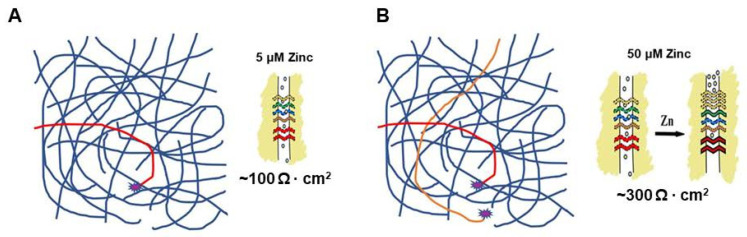
The 5 vs. 50 µM Zinc Level Effects on Downstream Regulation of Tight Junctions. Activation of Protein Kinase X by 5 µM Zinc (**A**) (red line) may give theoretical partial activation of a signaling network that results in the partial tightening of a TJ complex. At 50 µM zinc, however, (**B**), one would have activation of Protein Kinases X *and* Y (red and orange lines) that may give more complete activation of a signaling network, resulting in a more robust (less leaky) TJ complex. This scenario can hold if X and Y are fully separate kinases (signaling pathways) or if X and Y genes are two polymorphic alleles of the same kinase whose total activity falls dramatically at 5 µM zinc levels due to low activity from the Y variant.

**Figure 5 ijms-25-03452-f005:**
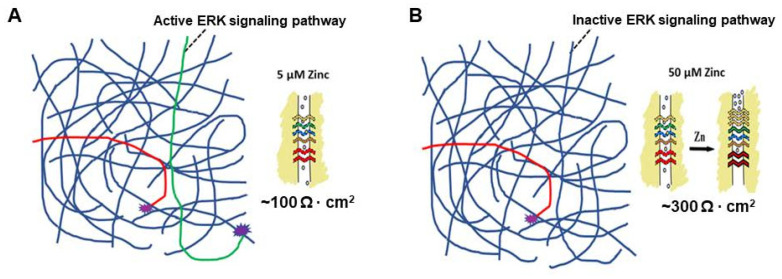
50 µM Zinc Resulting in Inhibition of a Signaling Pathway. Elevated levels of micronutrients need not simply result in the activation of signal transduction pathways. They could just as easily cause inhibition of a pathway that could then result in TJ remodeling and reduced leak. The reports of reduced pERK levels in airway and intestinal epithelia exposed to elevated levels of quercetin and zinc, respectively, may serve as an actual example of such regulation [59,68]. Panels (**A**) and (**B**) show a hypothetical example where elevated zinc levels inhibit an active ERK signaling pathway (green line), resulting in increased transepithelial electrical resistance/barrier function, an actual observation reported by [68].

## Data Availability

Not applicable.

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
