# Peer review of "Micronutrients at Supplemental Levels, Tight Junctions and Epithelial Barrier Function: A Narrative Review"

_ijms, 2024, doi:10.3390/ijms25063452_

Round 1
Reviewer 1 Report
Comments and Suggestions for Authors
Dear Authors,
Thank you very much for submitting your paper to IJMS. Below I present some minor/major suggestions:
- abstract - you mention 'this review' - I would recommend specifying it by adding 'narrative review'
- Figure 1 - I would recommend to provide a better quality of the picture because the font is a little bit blurred
- 'the entire GI tract' - I am aware that this abbreviation is pretty obvious but please add the full name before introducing any abbreviation
- please change the reference and citation style according to the guidelines of the IJMS
- Similarly to figure 1, figure 2 and 3 require correction since it is a little bit blurred and hard to read from it
- I would recommend adding a separate paragraph (at the very beginning of the paper) with information about the search strategy that was made in order to write this narrative review along with the objectives of your paper
- Please introduce the full names before introducing any abbreviations. Correct it in the whole paper. I know that TJ abbreviation have been already introduced in the abstract section however it it was not in the main text. So please provide the full name of it before adding the abbreviation
- Please remove the additional spaces between words - there are some of these in the whole text
Regards
Comments on the Quality of English LanguageSome minor grammatical mistakes require correction
Author Response
Please see the attachment called "Reply to 1st Reviewer"

Reviewer 2 Report
Comments and Suggestions for Authors
This manuscript is a review of roles for micronutrients as potential therapeutic adjuvants by improving barrier function which is otherwise compromised in diseases. This is an important topic and has clinical relevance. The authors have written a comparable review, "Micronutrient Improvement of Epithelial Barrier Function in Various Disease States: A Case for Adjuvant Therapy" ( Reference 22 ) that appeared in an IJMS collection "State-of-the-Art Bioactives and Nutraceuticals in USA" that covers a lot of similar ground. This includes a focus on how vitamin A, vitamin D and zinc can regulate tight junctions. Figures 1, 3 and 4 are also represented in the previous review article and several other points of discussion are present in both the published article and the current manuscript.
Where Reference 22 and the current manuscript differ, is that the published review has considerably more detail related to the molecular regulation of tight junctions, which is surprising considering this manuscript submission is for the collection "The Tight Junction and Its Proteins: From Structure to Pathologies". The current manuscript needs to have significantly more comprehensive coverage of how micronutrients impact specific signaling pathways and tight junction proteins. This is a critical concern that needs to be addressed by significant additions to the text.
The focus on zinc could also be expanded, e.g. how it is handled by cells and how zinc affects other signaling pathways beyond PKC reaction kinetics could be considered. As just one example, zinc dependent transcription factors could have an impact on tight junctions. How zinc is handled by cells could also be expanded in the manuscript and the impact that has on cytosolic zinc levels could be included in the manuscript
Figure 1 is misleading and over simplified in that it makes it appear that only a single layer of cells is responsible for tissue barrier function. For instance, this model neglects a role for the vascular endothelium and other barrier forming cells found in tissues. Please replace with a more detailed figure or remove it.
Figure 2 and Figure 6 depict tight junctions that make it appear that individual tight junction forming proteins are inserted into the plasma membrane of two cells across a cell-to-cell contact site. This is misleading and should be modified.
Figure 3 is redundant to Figure 4 and should be removed.
The tangled signaling pathway diagrams in Figure 6 are confusing and tough to follow. These diagrams should be replaced with real world examples of parallel pathways that could be regulated by differential zinc binding affinity. At minimum, an easier to understand model figure is needed.
"performing thermodynamic work". This concept is vague and readers could benefit from more detail. I assume this includes using energy to drive flux against a concentration gradient. Are there other aspects related to this concept that the authors were considering?
"The lynchpin in this entire scenario is the Tight Junctional (TJ) complex. As described in several review articles the TJ complex is just that - complex. Thirty integral membrane proteins can form the barrier itself, most notable being the 27-member (23 gene) claudin family. But in addition, there are approximately 15 TJ-associated, intracellular proteins - both structural and signaling proteins - that regulate the complex and interface it with the cytoskeleton." More detail is needed. Which proteins? Also, this does not take into account how different aspects of the apical junctional complex regulate tissue barrier function, which are likely to be sensitive to micronutrients. Where did the numbers 30 integral membrane proteins and ~15 TJ-associated, intracellular proteins come from?
"100,000 years ago, the human race was not nearly as mobile as today. If one happened to live in a certain valley where life was reasonably good and food was plentiful, one tended to stay put. That also meant that one's diet would be somewhat monotonous. And the monotony might have ensured high levels of specific micronutrients deriving from that diet. If one lived near a salmon-filled lake or stream, one ate a lot of salmon. If that valley also happened to contain plentiful supplies of mushrooms, one would consume a lot of those as well. And a diet high in certain mushrooms and salmon would make for a diet very high in zinc and Vitamin D. And thus, when one encountered a disease situation calling for enhanced barrier function (deriving from these micronutrients) one's epithelial cell layers would be ready. However, remove oneself from that high and chronic intake of specific micronutrients, and Protein Kinase Y's high Km shortfalls would become very apparent. And the need for micronutrient supplementation in specific disease situations would be born." Some citations to support this part of the discussion would be appreciated.
"Too often in medical research we seek (using a baseball analogy) to hit the home run. We seek how a disease can be cured or thoroughly prevented, as e.g., Salk, Sabin, and polio. Indeed, we witnessed this in the very strong focus on the mRNA vaccines during the COVID-19 pandemic. But (again using the baseball analogy), there is a very strong case to made for "small ball" in dealing with health crises. Hitting "singles" and "doubles" can be pivotally valuable in winning a game, and we would advance that micronutrient effects on barrier function can be just that." Please consider using a different analogy. For an international audience a baseball analogy is not straight forward.
Author Response
Please see the attachment called "Reply to 2nd Reviewer"

Round 2
Reviewer 1 Report
Comments and Suggestions for Authors
Dear Authors,
Thank you very much for providing an updated version of your manuscript.
- Please correct the quality/design of Figure 1. In this form, it does not look very professional and needs corrections
- The quality of Figure 2 should be corrected. It is very blurred and hard to read.
- please correct the reference style according to the requirements of the journal
Kind regards
Reviewer 2 Report
Comments and Suggestions for Authors
The authors have addressed the critiques which improved the manuscript. Considering the overall scope of the manuscript as a narrative review was helpful, as were the additions to Figures 6 and 7.
I still think that Figure 1 should be removed. Regarding the comment "Furthermore, going all the way back to Farquhar and Palade (1963), it’s been very broadly accepted that the epithelium with its Tight Junctional complexes is the principal component of barrier function in any epithelial tissue." I would argue that this depends on the vascular bed being examined. For instance the endothelium plays a major role in the blood brain barrier and the microvasculature is also a key component of overall lung barrier function. The concept of epithelial barrier function is better covered in the other figures in the manuscript as opposed to the diagram in Figure 1.
Regarding figure resolution/formatting issues that are persisting, I am pretty sure that these can be addressed during article production, but this should be confirmed.
